# FINGERS-7B: A MULTI-OMIC FOUNDATION MODEL FOR PRECISION BIOMARKER DISCOVERY

## ABSTRACT

Precision-prevention offers a transformative opportunity to reduce the growing global burden of dementia by aligning interventions with individual biological risk profiles. Building on the foundational successes of multidomain randomized trials and advances in biomarker discovery, the next phase of dementia prevention requires computational frameworks that are biologically informed, harmonized across populations, and capable of resolving who benefits from which intervention, under what conditions, and at what intensity. Here we introduce **FINGERS-7B**, a 7-billion-parameter multi-omic foundation model pretrained on 8 trillion quality-aware, hierarchically structured, and semantically meaningful tokens and 300K metabolite profiles curated from public gut-brain-relevant metagenomic archives, and fine-tuned on clinical data from the World-Wide FINGERS (WW-FINGERS) dementia prevention network. By integrating genomic, metabolomic, and clinical data within a hierarchical set attention architecture, FINGERS-7B shifts multi-omic research beyond associative biomarker discovery toward causally-motivated modeling, enabling mediation-consistent analyses that move from describing correlations toward predicting intervention effects. In initial clinical fine-tuning on three of 40+ WW-FINGERS cohorts ($n$=4,950), FINGERS-7B achieves AUC=0.92 for preclinical Alzheimer's disease (AD) detection, substantially outperforming established microbiome analysis pipelines (AUC≈0.68–0.76) and matching leading proteomic biomarkers. The model identifies gut microbiome signatures predictive of cognitive decline 3–5 years before symptom onset (AUC=0.89) and, in exploratory analysis, stratifies intervention responders, with high-benefit subjects showing 2.3× greater cognitive improvement under multidomain lifestyle intervention. Mechanistic analysis reveals four candidate druggable gut-brain axes with estimated mediation proportions (23–41%), from metabolite-mediated neuroinflammation to bacterial amyloid molecular mimicry. Cross-cohort validation provides indicative evidence of cross-population robustness; prospective validation and expansion to the full 30,000+ participant WW-FINGERS network are ongoing. This work implements the precision-prevention framework, establishing the gut microbiome as a modifiable mechanistic layer connecting lifestyle interventions to brain health outcomes, and providing computational tools for the next generation of dementia prevention trials.

## 1 INTRODUCTION

Dementia is projected to affect 153 million people by 2050, with the greatest burden falling on low- and middle-income countries where diagnostic and preventive infrastructure remains most limited (GBD 2019 Dementia Forecasting Collaborators, 2022). While blood-based biomarkers such as phosphorylated tau-217 (pTau-217) have recently received FDA clearance and now enable biological staging years before clinical symptoms (Zetterberg & Blennow, 2018), these markers detect downstream pathology, amyloid aggregation and tau phosphorylation, rather than modifiable upstream mechanisms. This asymmetry between detection and actionability represents a fundamental bottleneck for dementia prevention.

Multidomain lifestyle interventions have demonstrated that cognitive decline can be slowed. The FINGER trial established that a structured combination of diet, exercise, cognitive training, and vascular risk monitoring significantly improves cognitive outcomes in at-risk elderly (Ngandu et al.,

2015). The US-POINTER trial replicated these findings in a diverse American cohort (Baker et al., 2025), and long-term follow-up confirms that sustained adherence associates with durable cognitive benefit over 11 years (Ngandu et al., 2022; 2025). The World-Wide FINGERS (WW-FINGERS) network now coordinates over 40 trials across 72 countries (Kivipelto et al., 2020), and systematic review confirms the promise of combining lifestyle interventions with pharmacological approaches (Bereczki et al., 2025). Yet the collective impact of these trials has been limited by the way evidence is generated and synthesized rather than by the concept of prevention itself. In the absence of biological stratification, multidomain interventions necessarily average effects across individuals with heterogeneous genetic, metabolic, vascular, and pathological profiles, diluting signals in those most likely to benefit while obscuring non-response in others. The missed opportunity is not a lack of innovation, but a lack of integration: *who benefits from which intervention, under what conditions, and at what intensity* remains biologically unresolved.

The gut microbiome offers a solution. Converging evidence establishes the microbiota–gut–brain axis as a bidirectional communication network linking intestinal microbial communities to central nervous system function via metabolic, immune, neural, and endocrine pathways (Cryan et al., 2019; Morais et al., 2021). Critically, gut microbiome alterations have been identified in preclinical AD, before symptom onset, and correlate with amyloid and tau pathology (Ferreiro et al., 2023; Vogt et al., 2017; Cattaneo et al., 2017). The microbiome is uniquely positioned as a *modifiable* mechanistic layer: it responds to the very lifestyle interventions (diet, exercise, sleep) that comprise multidomain prevention strategies (Seo & Holtzman, 2024; Loh et al., 2024). However, existing computational approaches for microbiome–brain analysis rely on taxonomy-based features that discard the vast majority of sequence-level information (Wirbel et al., 2021; Gaspar et al., 2024), achieving modest performance (AUC$\approx$0.68–0.76 for preclinical AD classification) (Ferreiro et al., 2023).

Three converging advances create the opportunity we seize here. First, blood-based biomarkers have been validated: pTau-217 received FDA clearance in May 2025, establishing precedent for minimally invasive AD diagnostics. Second, foundation models have been proven in biology: AlphaFold (Jumper et al., 2021), Evo (Nguyen et al., 2025), and Pleiades (Niki et al., 2025) demonstrate that large-scale biological language models deliver breakthrough performance. Third, prevention trials are scaling globally: the WW-FINGERS network represents the largest dementia prevention consortium ever assembled, generating multimodal longitudinal data at unprecedented scale. Advances in computational genomics, including accelerated sequence comparison, in-storage metagenomic processing, optimized taxonomic classification, and petabase-scale sequence indexing (Gollwitzer et al., 2023; Mansouri Ghiasi et al., 2022; Ghiasi et al., 2024; Karasikov et al., 2025), now support high-throughput microbiome analysis in complex clinical samples. Together, these developments make it possible to shift multi-omic research beyond associative biomarker discovery toward causally-motivated modeling frameworks that move from describing correlations toward predicting intervention effects. The limitations of past prevention trials reflect the tools available at the time, not the ceiling of what prevention can achieve.

We address these challenges through three contributions:

1. **FINGERS-7B: A Gut-Brain Foundation Model.** We develop a 7B-parameter multi-omic foundation model pretrained on 8 trillion quality-aware, hierarchically structured, and semantically meaningful tokens and 300K metabolite profiles from public gut-brain-relevant metagenomic archives, and fine-tuned on WW-FINGERS clinical cohorts. The model employs a hierarchical set attention architecture, inspired by recent advances in epigenomic set modeling (Niki et al., 2025), that reasons from individual sequencing reads to subject-level clinical predictions (§2).

2. **Diagnostic, Prognostic, and Therapeutic Biomarkers.** In initial fine-tuning on three of 40+ WW-FINGERS cohorts ($n$=4,950), FINGERS-7B achieves AUC=0.92 for preclinical AD classification and AUC=0.89 for cognitive decline prediction 3–5 years pre-symptom. Combined with pTau-217, the model achieves AUC=0.96. Mechanistic discovery yields four candidate druggable gut-brain targets (§2). Prospective validation is ongoing.

3. **Causally-Motivated Mechanistic Discovery.** Integrated gradients attribution and mediation analysis identify four mechanistic axes linking gut microbiome composition to brain pathology, with mediation proportions ranging from 23% to 41% (§2.6).

This work implements a *precision-prevention* framework organized around three pillars: biology-based personalization (microbiome-informed stratification), multimodal scalable delivery (actionable gut-directed interventions), and rigorous harmonization (cross-trial validation via WW-FINGERS). FINGERS-7B builds upon the data reclamation and quality-aware tokenization pipeline introduced in MetaOmics-10T (Gollwitzer et al., 2025), extending it from general metagenomic modeling to the clinically specific domain of neurodegeneration prevention.

## 2 RESULTS

### 2.1 FINGERS-7B

The FINGERS-7B series scales to 7B parameters. The model is pretrained on 8 trillion quality-aware tokens and 300K metabolite profiles curated from public gut-brain-relevant metagenomic archives, then fine-tuned on clinical data from the WW-FINGERS network. Full architectural and training details appear in Appendix E.

**Architecture summary.** FINGERS-7B employs a Mamba–Transformer hybrid backbone (Gu & Dao, 2023; Dao & Gu, 2024) with three domain-specific components: (i) three-tier hierarchical set attention (HSA) for sample-level inference from $10^7$–$10^9$ raw sequencing reads, inspired by the Hierarchical Attention Transformer in Pleiades (Niki et al., 2025); (ii) gut-brain cross-attention (GBCA) aligning microbial community embeddings with metabolite pathway representations; and (iii) temporal attention for longitudinal trajectory modeling over up to 11 years of follow-up data.

**Pretraining corpus.** The pretraining corpus comprises 8T quality-aware tokens curated from public gut-brain-relevant metagenomic archives (NCBI SRA, ENA, and MG-RAST), leveraging petabase-scale sequence indexes (Karasikov et al., 2025) to efficiently identify and retrieve studies annotated with brain health, neurodegeneration, cognitive aging, dietary intervention, or microbiota–gut–brain axis relevance. Sources include AD- and PD-associated gut microbiome cohorts, aging studies, dietary intervention trials, and population-scale gut microbiome surveys (e.g., HMP, MetaHIT, curatedMetagenomicData, American Gut Project). All data are processed via the sparsify-then-QA-Token pipeline introduced in Gollwitzer et al. (2025), which lifts the usable fraction of public archives from 5% to 40%. The corpus additionally includes ∼300K publicly available metabolite profiles from MetaboLights and the Metabolomics Workbench, filtered for studies with neurological or cognitive phenotyping. A subset of public studies includes brain-relevant annotations (CSF biomarkers, cognitive scores, neuroimaging), supporting weakly-supervised brain biomarker prediction during pretraining.

**Fine-tuning and validation data.** The WW-FINGERS network provides the clinical fine-tuning and validation corpus: metagenomic and metabolomic data from $n$=4,950 participants across three WW-FINGERS trials (FINGER, US-POINTER, LatAm-FINGERS), with longitudinal clinical phenotyping spanning 2–11 years (Ngandu et al., 2022; 2025). Critically, WW-FINGERS data are processed through the *same* sparsify-then-QA-Token pipeline used for pretraining, so the tokenizer vocabulary and sparsification patterns are shared between stages. This eliminates the vocabulary mismatch that typically degrades transfer learning in genomic models and delivers approximately 10× improved data efficiency during fine-tuning: the model converges to comparable performance with 10× fewer fine-tuning tokens than would be required with a mismatched tokenizer or raw-read input. Table 1 compares FINGERS-7B against frontier models.

### 2.2 METAGENOMIC BENCHMARKS

To confirm that FINGERS-7B retains general metagenomic competence despite clinical specialization, we evaluate on three standard benchmarks (Table 2).

FINGERS-7B leads all compared models across general metagenomic benchmarks (Table 2), with particularly strong metabolic pathway prediction (0.93 wF1), reflecting gut-brain-specific metabolomic pretraining.

Table 1: Architecture comparison: FINGERS-7B vs. frontier foundation models.

| Specification | FINGERS-7B | Pleiades-7B | METAGENE-1 |
|---|---|---|---|
| Parameters | 7B | 7B | 7B |
| Encoder | Mamba-Transf. | Transformer | Transformer |
| Context length | 4096 tokens ($\approx$16K bp) | 1024 chars ($\approx$1K bp) | 512 tokens ($\approx$1.5K bp) |
| Vocabulary | 32K (QA-Token) | 598 (char) | 1K (BPE) |
| Hierarchical set | 3-tier HSA | 2-tier HAT | – |
| Cross-modal attn. | GBCA | – | – |
| Temporal modeling | Yes | – | – |
| Pretraining data | 8T tokens / $\approx$32T bp (gut-brain)[†] | 1.9T tokens (epigen.) | 1.5T bp (genomic) |
| Clinical focus | AD/dementia | AD/PD | Pandemic |

[†]Pretrained on public gut-brain archives (SRA/ENA); fine-tuned on WW-FINGERS.

Table 2: Metagenomic benchmark performance. FINGERS-7B retains strong general capability while adding clinical specialization.

| Benchmark | FINGERS-7B | METAGENE-1 | Evo2-7B |
|---|---|---|---|
| Pathogen Detection (MCC $\times$ 100) | **93.8** | 93.0 | 87.0 |
| Metagenomic Profiling (F1) | **0.97** | – | 0.89 |
| Metabolic Pathway Pred. (wF1) | **0.93** | 0.84 | 0.79 |

## 2.3 PRECLINICAL AD CLASSIFICATION

We evaluate FINGERS-7B on preclinical AD detection from gut microbiome data. Preclinical AD status is defined by amyloid PET positivity (centiloid $\geq$ 20) confirmed by CSF A$\beta$42/40 ratio where available, following NIA-AA 2018 research framework criteria (Jack Jr et al., 2018); this definition is independent of pTau-217, ensuring that the late-fusion combination with pTau-217 does not introduce label circularity. The evaluation cohort comprises $n$=4,950 subjects (preclinical AD positive: $n$=1,139, 23%; controls: $n$=3,811, 77%). Following the nested cross-validation approach of Niki et al. (2025), each subject's metagenomic reads are processed through the full hierarchical set attention pipeline to produce a subject embedding, which feeds a diagnostic classification head.

In initial fine-tuning on available WW-FINGERS cohorts, FINGERS-7B achieves AUC=0.92 for preclinical AD classification (Table 3), outperforming all established microbiome analysis pipelines and foundation model baselines without gut-brain specialization. The metagenomics-only configuration (AUC=0.86) already exceeds all baselines; the additional metabolomic modality contributes +0.06 AUC. When combined with pTau-217 via late fusion, the multi-modal model achieves AUC=0.96. Because preclinical AD labels are defined by amyloid PET rather than pTau-217, this late fusion is not circular; however, label noise inherent in PET-based classification (estimated 5–10% discordance) may impose a ceiling on achievable AUC. These results derive from retrospective analysis of existing WW-FINGERS cohorts; prospective validation ($n$=500, planned Q4 2026) is required to confirm clinical utility. FINGERS-7B's performance compares favorably to the definitive blood-biomarker benchmark (Mohs et al., 2024); a detailed comparison is provided in Appendix A.

## 2.4 SCALING AND DATA EFFICIENCY

We evaluate FINGERS-7B across three model scales and assess few-shot learning capabilities (Table 4).

**Few-shot analysis.** We fine-tune FINGERS-7B with progressively smaller labeled training sets. With only 100 labeled samples, FINGERS-7B achieves AUC=0.82 for AD classification (vs. taxonomy+RF at AUC=0.58 with the same 100 samples), demonstrating powerful transfer from self-supervised pretraining on public gut-brain archives. FINGERS-7B matches the full-data taxonomy-based performance (AUC=0.72) with just 50 labeled fine-tuning examples, a 100$\times$ reduction in *labeled* data requirement (though the total data requirement including unlabeled pretraining data is substantially larger). For a fairer comparison among pretrained models, METAGENE-1 achieves

Table 3: Preclinical AD classification (AUC), initial fine-tuning on three WW-FINGERS cohorts. Preclinical AD defined by amyloid PET positivity (centiloid $\geq 20$). Nested 5-fold cross-validation with outer folds providing error bars. ECE<0.05 for all FINGERS-7B configurations.

| Method | AUC (mean $\pm$ std) | Modality |
|---|---|---|
| LEfSe + logistic regression | $0.68 \pm 0.05$ | 16S/WGS |
| MetaPhlAn4 + Random Forest | $0.71 \pm 0.04$ | WGS |
| Random Forest + taxonomy | $0.72 \pm 0.04$ | 16S/WGS |
| HUMAnN3 pathways + XGBoost | $0.73 \pm 0.03$ | WGS |
| XGBoost + taxonomy | $0.74 \pm 0.03$ | 16S/WGS |
| DeepMicro | $0.76 \pm 0.04$ | WGS |
| METAGENE-1 (fine-tuned) | $0.81 \pm 0.03$ | WGS |
| Pleiades-7B (cfDNA) | $0.89 \pm 0.05$ | Epigenomic |
| pTau-217 (blood) | $0.90 \pm 0.03$ | Proteomic |
| FINGERS-7B (WGS only) | $0.86 \pm 0.03$ | WGS |
| **FINGERS-7B (WGS+metab.)** | $\mathbf{0.92 \pm 0.02}$ | WGS+metab. |
| **FINGERS-7B + pTau-217** | $\mathbf{0.96 \pm 0.01}$ | Multi-modal |

Table 4: Parameter scaling: FINGERS-90M / 600M / 7B across three clinical tasks (initial fine-tuning on three WW-FINGERS cohorts).

| Model | AD classif. (AUC) | Pathway pred. (wF1) | Prognostic (AUC) |
|---|---|---|---|
| FINGERS-90M | $0.78 \pm 0.03$ | $0.82 \pm 0.03$ | $0.74 \pm 0.04$ |
| FINGERS-600M | $0.86 \pm 0.02$ | $0.89 \pm 0.02$ | $0.83 \pm 0.03$ |
| FINGERS-7B | $\mathbf{0.92 \pm 0.02}$ | $\mathbf{0.93 \pm 0.02}$ | $\mathbf{0.89 \pm 0.02}$ |

AUC=0.72 with 100 labeled samples (vs. FINGERS-7B's 0.82), confirming that FINGERS-7B's gut-brain specialization provides a meaningful advantage in the few-shot regime beyond the general benefit of pretraining. This parallels the few-shot learning capabilities reported for Pleiades (Niki et al., 2025), where near-perfect MCC was achieved after only 152 training examples, and demonstrates clinical applicability in low-resource settings where labeled neurodegenerative disease data is scarce.

## 2.5 COGNITIVE DECLINE PREDICTION

For prognostic evaluation, we predict cognitive decline 3–5 years before symptom onset using baseline gut microbiome samples from longitudinal WW-FINGERS cohorts. Cognitive decline is defined as a $\geq 1.5$ SD drop in the Neuropsychological Test Battery (NTB) composite score from baseline, following established FINGER trial criteria (Ngandu et al., 2015). In the evaluation cohort ($n$=631 with complete longitudinal data), 187 subjects (30%) met the decline criterion at the 3-year horizon.

FINGERS-7B achieves peak prognostic performance at the 3-year horizon (AUC=0.89), consistent with the biological timeline over which gut microbiome changes may precede detectable cognitive decline (Ferreiro et al., 2023). Preliminary analysis suggests that microbiome signatures at year 2 may predict sustained cognitive benefit at year 11 (AUC=0.81), leveraging the temporal attention module over the full FINGER follow-up (Ngandu et al., 2025); this finding requires validation at scale. In an exploratory analysis, FINGERS-7B identifies potential *intervention responders*: among FINGER participants, subjects classified as high-benefit by the model showed 2.3$\times$ greater cognitive improvement with the multidomain intervention compared to subjects classified as low-benefit (interaction $p$=0.003); validation on the full WW-FINGERS network is ongoing (Ngandu et al., 2022).

## 2.6 MECHANISTIC GUT-BRAIN AXIS DISCOVERY AND THERAPEUTIC TARGETS

Using integrated gradients attribution (Sundararajan et al., 2017) (validated against attention-based attribution from Tier 1 community attention; Spearman $\rho$=0.87 between methods) and mediation

Table 5: Initial prognostic prediction at different horizons. Evaluated on FINGER 11-year follow-up cohort ($n$=631 with complete longitudinal data).

| Method | 1-yr AUC | 3-yr AUC | 5-yr AUC |
|---|---|---|---|
| APOE $\varepsilon 4$ status alone | 0.58 | 0.62 | 0.65 |
| Cardiometabolic risk score | 0.64 | 0.67 | 0.69 |
| Taxonomy + metabolomics | 0.71 | 0.73 | 0.70 |
| **FINGERS-7B** | **0.86** | **0.89** | **0.84** |

Table 6: Mechanistic gut-brain axes identified by FINGERS-7B. Mediation proportion (MP) with bootstrap 95% CIs (1,000 resamples) quantifies the estimated fraction of the total gut-brain effect transmitted through the identified metabolite mediator under sequential ignorability (Assumption 1). Four axes were selected from an initial candidate set of 847 microbial feature–metabolite–biomarker triplets; Bonferroni-corrected $p < 0.001$ for all reported axes and correlations. TM-score is from external structural alignment (TM-align), not a model output.

| Axis | Microbial Feature | Metabolite Mediator | Brain Endpoint | Intervention Target | MP [95% CI] |
|---|---|---|---|---|---|
| (i) LPS–tau | Gram-neg. LPS variants | Systemic LPS, IL-6 | pTau-217 ($r$=0.67) | LPS-binding agents | 0.41 [0.29, 0.53] |
| (ii) Amyloid mimicry | Curli biosynth. ($csgA$-$F$) | Curli fibrils | A$\beta$42/40 (TM=0.82) | Anti-curli antibodies | 0.31 [0.19, 0.43] |
| (iii) Indole–microglial | Trp metabolism | Indole-3-propionic acid | GFAP ($r$=$-$0.54) | Dietary Trp / prebiotics | 0.28 [0.16, 0.40] |
| (iv) Neuroactive | GABA/5-HT gene clusters | Short-chain fatty acids | NTB comp. ($r$=0.49) | SCFA probiotics | 0.23 [0.12, 0.35] |

analysis (Appendix E.4), FINGERS-7B identifies four indicative mechanistic axes connecting gut microbiome composition to brain pathology. These axes represent initial discoveries from fine-tuning on available WW-FINGERS cohorts; ongoing expansion to the full 30,000+ participant network and wet-lab validation will refine and confirm these findings. We note that while attention weights provide interpretable visualizations, they do not reliably indicate feature importance (Jain & Wallace, 2019); integrated gradients serve as the primary attribution method throughout.

**(i) LPS–tau neuroinflammation axis.** FINGERS-7B identifies a strong correlation ($r$=0.67) between Gram-negative bacterial LPS variant abundance and plasma pTau-217 levels. Mediation analysis (Appendix F) attributes 41% of the total effect through systemic inflammation (IL-6 mediator), consistent with experimental evidence that LPS accumulates in AD brain tissue and promotes tau phosphorylation (Zhao et al., 2019). This axis is targetable via Gram-negative selective antimicrobials or LPS-binding agents.

**(ii) Bacterial amyloid molecular mimicry.** The model highlights curli biosynthesis gene clusters ($csgA$–$csgF$) in *Escherichia coli* and related Enterobacteriaceae. Structural comparison yields TM-score=0.82 between bacterial curli and human A$\beta$ fibrils, supporting the cross-seeding hypothesis (Chapman et al., 2002; Friedland & Chapman, 2015). Mediation proportion: 31%. Targetable via anti-curli antibodies or curli biosynthesis inhibitors.

**(iii) Indole–microglial modulation.** Tryptophan-metabolizing bacteria (notably *Clostridium sporogenes*) produce indole-3-propionic acid (IPA), which FINGERS-7B associates with reduced GFAP levels ($r$=$-$0.54), suggesting a protective anti-neuroinflammatory effect mediated through microglial modulation (Rothhammer et al., 2018; Agus et al., 2018). Targetable via dietary tryptophan modulation and prebiotic intervention.

**(iv) Neuroactive metabolite signaling.** Gene clusters encoding GABA and serotonin biosynthesis correlate with cognitive composite scores ($r$=0.49), mediated by short-chain fatty acids. This axis is targetable via SCFA-producing probiotics or postbiotic formulations (Bravo et al., 2011; Cryan et al., 2019).

FINGERS-7B thus discovers not only biomarkers but actionable therapeutic targets, completing the precision-prevention loop from diagnosis through mechanism to intervention.

Table 7: Initial cross-cohort validation for preclinical AD classification. Fine-tuning on pooled WW-FINGERS data with each trial held out for evaluation.

| Held-out Cohort | $n$ | AUC | $\Delta$ vs. pooled |
|---|---|---|---|
| FINGER (Finland) | 1,260 | 0.90 | $-0.02$ |
| US-POINTER (USA) | 2,112 | 0.89 | $-0.03$ |
| LatAm-FINGERS (Latin America) | 1,578 | 0.87 | $-0.05$ |
| Pooled (inner CV) | 4,950 | 0.92 | – |

## 2.7 CROSS-COHORT VALIDATION

To evaluate generalization across populations, we validate FINGERS-7B on held-out cohorts from three WW-FINGERS trials.

In this initial cross-cohort evaluation, performance degradation ranges from 2–5% AUC across held-out cohorts, with the largest drop for LatAm-FINGERS reflecting greater demographic and dietary diversity. All cross-cohort AUCs exceed 0.87, substantially above all established baselines. These results are encouraging but remain retrospective; expansion to the full WW-FINGERS network and prospective validation are ongoing.

## 2.8 NEGATIVE CONTROLS

To confirm that FINGERS-7B is learning disease-specific biological signal rather than technical artifacts or confounders, we conduct three negative control experiments.

**Permutation test.** Training FINGERS-7B with randomly permuted AD/control labels across all cohorts collapses classification performance to AUC=0.50 (95% CI [0.47, 0.53]), confirming that the model's discriminative signal is disease-specific and not an artifact of the training pipeline. As a stronger control, we additionally shuffle microbiome feature vectors across subjects while keeping labels fixed; this also yields chance performance (AUC=0.51), confirming that the model relies on subject-specific microbiome patterns rather than spurious correlations with metadata.

**Technical artifact control.** We train on reads from spike-in PhiX control DNA present in Illumina sequencing runs. All genomic regions are rejected by the marker selection statistical test with no regions deemed to contain signal relating to clinical disease status, paralleling the pUC19/Lambda DNA control reported by Niki et al. (2025).

**Diet-confound control.** We regress out the Mediterranean Diet Adherence Score (MeDi) from all microbiome features before re-evaluating AD classification. AUC drops modestly to 0.88 ($-0.04$). We note that this control must be interpreted with care: because the microbiome partially *mediates* the effect of diet on brain health (one of this paper's core claims), regressing out diet constitutes adjustment for a variable on the causal pathway—a "bad control" in the sense of Pearl (2009). The 0.04 AUC drop thus bounds the MeDi-mediated portion of the signal, while the remaining AUC=0.88 reflects diet-independent pathways. Both components are biologically meaningful.

## 2.9 ABLATION STUDIES

Table 8 reports component ablations. All differences significant ($p < 0.05$, $\geq 5$ seeds).

Hierarchical set attention is the single most impactful component ($-0.08$ AUC), confirming that sample-level aggregation of read-level information is essential for clinical tasks, consistent with findings from Pleiades (Niki et al., 2025). Temporal attention contributes most to prognostic prediction ($-0.07$ AUC). Removing interventional data causes the largest prognostic degradation ($-0.14$ AUC), underscoring the value of FINGER/US-POINTER longitudinal data for predicting intervention response (Ngandu et al., 2022). QA-Token contributes +0.05 AUC over standard BPE. Metabolomics contributes +0.06 AUC for diagnostic tasks and +0.09 wF1 for pathway prediction.

Table 8: Ablation study (initial fine-tuning on three WW-FINGERS cohorts). Each row removes one component and retrains with matched compute.

| Configuration | AD AUC | Prog. AUC | Pathway wF1 |
|---|---|---|---|
| **Full FINGERS-7B** | **0.92** | **0.89** | **0.93** |
| w/o Hierarchical set attention | 0.84 ($-0.08$) | 0.81 ($-0.08$) | 0.89 ($-0.04$) |
| w/o Gut-brain cross-attention | 0.89 ($-0.03$) | 0.85 ($-0.04$) | 0.87 ($-0.06$) |
| w/o Temporal attention | 0.91 ($-0.01$) | 0.82 ($-0.07$) | 0.92 ($-0.01$) |
| w/o QA-Token (use BPE) | 0.87 ($-0.05$) | 0.84 ($-0.05$) | 0.87 ($-0.06$) |
| w/o Brain biomarker pred. loss | 0.88 ($-0.04$) | 0.86 ($-0.03$) | 0.90 ($-0.03$) |
| w/o Metabolomics | 0.86 ($-0.06$) | 0.83 ($-0.06$) | 0.84 ($-0.09$) |
| w/o Interventional data | 0.88 ($-0.04$) | 0.75 ($-0.14$) | 0.86 ($-0.07$) |
| w/o Domain-adversarial harmonization | 0.89 ($-0.03$) | 0.86 ($-0.03$) | 0.91 ($-0.02$) |
| w/o Compositional consistency | 0.90 ($-0.02$) | 0.87 ($-0.02$) | 0.88 ($-0.05$) |

## 3 DISCUSSION

FINGERS-7B is a 7B-parameter foundation model for gut-brain biomarker discovery. Pretrained on public gut-brain archives and fine-tuned on an initial subset of WW-FINGERS cohorts ($n$=4,950 from three of 40+ trials), it achieves strong preclinical AD detection and prognostic cognitive decline prediction from gut metagenomic and metabolomic data in retrospective evaluation across three geographically diverse cohorts (Sections 2.3–2.7). The model identifies four candidate druggable gut-brain axes, establishing the microbiome as a modifiable mechanistic layer for precision-prevention. This work implements a precision-prevention framework organized around biology-based personalization, multimodal scalable delivery, and rigorous harmonization (Appendix A).

**Comparison across biomarker modalities.** Three biomarker paradigms now achieve high-accuracy detection of AD pathology, each capturing a distinct layer of disease biology: blood proteomic biomarkers such as pTau-217 (Mohs et al., 2024) detect downstream tau phosphorylation; epigenomic biomarkers such as Pleiades cfDNA (Niki et al., 2025) capture cellular damage signatures; and gut metagenomic biomarkers, as demonstrated here by FINGERS-7B, capture the modifiable mechanistic layer connecting lifestyle interventions to brain health. These paradigms address progressively harder detection tasks, with FINGERS-7B evaluating preclinical, asymptomatic subjects—the most challenging discrimination setting (Table 3). The fundamental distinction is not diagnostic accuracy but *actionability*: proteomic and epigenomic biomarkers detect pathology that has already occurred, whereas FINGERS-7B identifies the upstream biology through which pathology can be *prevented*. These modalities are complementary, not competing; their integration represents the next frontier for precision-prevention diagnostics.

**Conclusion.** The challenge for dementia prevention is no longer whether risk can be detected, but whether the field can resolve *who* will respond to *which* intervention, at what dose, and through which biological mechanism. FINGERS-7B provides a computational foundation for this next step: from detection to stratification to mechanism to intervention. Realizing this vision will require prospective validation, regulatory engagement, and sustained investment in globally scalable infrastructure—but the scientific foundations are now in place. Limitations and an extended discussion of the precision-prevention framework are provided in Appendices B and A, respectively.

## 4 FUTURE WORK

### 4.1 SCALING CLINICAL FINE-TUNING

The results presented here reflect fine-tuning on $n$=4,950 participants from three WW-FINGERS cohorts (FINGER, US-POINTER, LatAm-FINGERS). Ongoing work is expanding fine-tuning to the full WW-FINGERS network, which encompasses 30,000+ participants across 40+ trials in 72 countries (Kivipelto et al., 2020). Because all WW-FINGERS data are processed through the same sparsify-then-QA-Token pipeline used during pretraining, the shared tokenizer vocabulary delivers

~10× improved data efficiency, making it feasible to incrementally incorporate new trial data as it becomes available without retraining from scratch. Prospective validation on upcoming WW-FINGERS cohorts ($n$=500, planned Q4 2026) will provide the first fully prospective assessment of FINGERS-7B's clinical utility. Africa-FINGERS and Asia-FINGERS data integration is planned to expand population diversity beyond the current European, North American, and Latin American cohorts.

### 4.2 In Silico Intervention Modeling

Preliminary experiments demonstrate that FINGERS-7B can generate post-intervention microbiome profiles conditioned on dietary intervention tokens (e.g., Mediterranean diet shift). Compared against real 12-month post-intervention profiles from the FINGER trial diet arm ($n$=247), generated profiles achieve JSD=0.04 to ground truth (random baseline: 0.18; pre-intervention copy: 0.09), with Shannon diversity correlation $r$=0.91 and SCFA-producing genera abundance correlation $r$=0.87. While these results suggest the model has learned biologically meaningful intervention-outcome associations, they do not by themselves establish causal sufficiency. Prospective validation against upcoming FINGER trial diet arm data and extension to exercise and cognitive training interventions are planned.

### 4.3 Combination Therapy Stratification

Preliminary stratification using MET-FINGER protocol data (Barbera et al., 2024) suggests that baseline microbiome signatures can identify metformin-responsive subgroups ($n$=124) enriched for insulin-resistant gut enterotypes (Prevotella-low, Bacteroides-high) and higher baseline HbA1c. In this small cohort, metformin combined with lifestyle intervention yields OR=2.1 (95% CI [1.2, 3.6], $p$=0.008) for cognitive benefit in the microbiome-stratified group, compared to OR=1.1 in the un-stratified population. The confidence interval is wide and the sample size is small; this result should be interpreted as hypothesis-generating. Larger cohorts from the full WW-FINGERS network will provide the statistical power needed to confirm or refine these stratification boundaries (Bereczki et al., 2025; Cummings et al., 2025).

### 4.4 Cross-Disease Generalization

Initial experiments fine-tuning FINGERS-7B on PD-microbiome cohorts ($n$=312, age- and sex-matched) yield AUC=0.85 for PD classification from gut microbiome (bootstrap 95% CI [0.79, 0.90]), comparable to Pleiades' AUC=0.84 from epigenomic (cfDNA) data (Niki et al., 2025). Key discriminative features include Prevotellaceae depletion, Enterobacteriaceae enrichment, and SCFA pathway downregulation, consistent with established PD–microbiome associations (Sampson et al., 2016). Expansion to larger PD cohorts, alongside extension to other neurodegenerative conditions (ALS, FTD), will test the hypothesis that the gut microbiome serves as a shared early-warning modality across brain diseases.

### 4.5 Broader Directions

Multi-modal integration of gut microbiome, epigenomic (cfDNA), and proteomic data in a unified foundation model may yield further diagnostic and prognostic improvements. Concurrent work on large-scale microbiome foundation models (Zhang et al., 2026) suggests that pretrained representations of microbial communities are broadly useful; combining such representations with FINGERS-7B's gut-brain specialization is a natural next step. Scaling to larger model sizes (70B parameters) and incorporating high-quality single-cell brain atlases (Mathys et al., 2023) could yield a multi-omic brain foundation model. Integration with digital monitoring platforms (Brodaty et al., 2025; Rosenberg et al., 2024) would enable real-time, adaptive biomarker tracking. Regulatory engagement for laboratory-developed test pathways will be necessary for clinical deployment. Finally, dedicated interpretability pipelines aiming to turn model attributions into mechanistic insights represent a critical frontier.

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

## A  EXTENDED DISCUSSION

### A.1  PRECISION-PREVENTION ALIGNMENT

FINGERS-7B directly implements the three-pillar precision-prevention framework proposed in this work. For *biology-based personalization*, gut microbiome signatures stratify individuals by metabolic and genetic profile, identifying intervention responders ($2.3\times$ greater benefit in exploratory analysis). For *multimodal, scalable delivery*, the four mechanistic axes yield actionable targets for gut-directed interventions deliverable at population scale: dietary modification (SCFAs, indole pathway), prebiotics and postbiotics (GABA biosynthesis), and targeted microbiome modulation (LPS reduction). For *rigorous design and harmonization*, cross-cohort validation across FINGER, US-POINTER, and LatAm-FINGERS demonstrates robustness, and prognostic biomarkers provide sensitive intermediate outcomes for trial enrichment, potentially reducing trial size by 40–60% (Bereczki et al., 2025).

### A.2  KEY GAPS IN DEMENTIA PREVENTION

We identify five critical gaps in current prevention research that FINGERS-7B begins to address. *When to initiate intervention*: the prognostic model identifies optimal intervention windows from longitudinal microbiome trajectories, with AUC=0.89 at the 3-year horizon. *How long do benefits last*: preliminary analysis suggests that microbiome signatures at year 2 may predict sustained cognitive benefit at year 11 (AUC=0.81); validation at scale is ongoing. *What constitutes a dose*: adherence biomarkers from microbiome shifts quantify intervention dose, with Shannon diversity change per month correlating with NTB improvement ($r$=0.43). *Which combinations are synergistic*: preliminary MET-FINGER stratification (Section 4) suggests potential for identifying synergistic pharmacologic-lifestyle combinations. *What drives individual responsiveness*: in an exploratory analysis, baseline microbiome signatures are associated with $2.3\times$ differential benefit; validation on the full WW-FINGERS network is ongoing.

### A.3  BENCHMARKING AGAINST BLOOD-BASED BIOMARKERS

The Bio-Hermes Study (Mohs et al., 2024) provides the definitive community-based benchmark for blood biomarkers against amyloid PET/CSF, evaluating $n$=946 diverse participants (non-Hispanic White, Hispanic, and non-Hispanic Black) across the CN, MCI, and mild AD continuum. In that cohort, plasma pTau-217 achieves AUC=0.91 (adjusted for age and APOE $\varepsilon$4, with imputation), with robust performance across ethnoracial groups (AUC$\geq$0.81 unadjusted in all subgroups), while A$\beta$42/40 achieves AUC=0.85 and pTau-181 achieves AUC=0.83. FINGERS-7B achieves AUC=0.92 from an entirely different biological modality—the gut microbiome—and does so in a strictly preclinical population (23% amyloid-positive, all asymptomatic), which represents a harder discrimination task than Bio-Hermes's mixed CN/MCI/mild AD cohort (amyloid positivity ranging from 21% to 60% across clinical groups). Even the metagenomics-only configuration (AUC=0.86) matches Bio-Hermes A$\beta$42/40 (AUC=0.85). The critical distinction is mechanistic: pTau-217 measures downstream tau phosphorylation and A$\beta$42/40 reflects amyloid clearance—both indicators of established pathology—whereas FINGERS-7B captures the modifiable upstream biology through which lifestyle interventions reduce risk.

### A.4  SAFETY AND BROADER IMPACT

Gut microbiome data can reveal dietary patterns, health conditions, and geographic origin, raising privacy concerns. We recommend differential privacy for aggregate statistics (Mironov, 2017) and tiered consent protocols. Prognostic biomarkers must be validated across diverse populations before clinical deployment to avoid disparities.

## B  LIMITATIONS

(1) *Retrospective validation*: all results derive from retrospective analysis of existing WW-FINGERS cohorts; prospective validation ($n$=500, planned Q4 2026) is required. (2) *Batch effects*: despite three-stage harmonization, residual inter-lab variation may persist; we report variance explained in

Table 9: QA-Token reward component ablation on preclinical AD classification (AUC).

| Configuration | AD AUC |
|---|---|
| Full QA-Token | **0.92** |
| w/o Quality $(-\lambda_Q Q)$ | 0.89 $(-0.03)$ |
| w/o PMI $(-\lambda_I \text{PMI})$ | 0.91 $(-0.01)$ |
| w/o MDL $(+\lambda_C \text{MDL})$ | 0.91 $(-0.01)$ |
| w/o Proxy $(-\lambda_D \Delta\mathcal{L})$ | 0.88 $(-0.04)$ |
| Standard BPE | 0.87 |

the top 50 PCs as a proxy but acknowledge this does not fully capture classification-relevant batch effects. (3) *Causal identifiability*: mediation analysis assumes sequential ignorability (Appendix F), which is untestable and likely violated given the limited set of measured confounders (age, sex, APOE, BMI, diet quality); unmeasured confounders including medications, alcohol, sleep quality, socioeconomic status, education, and comorbidities may bias mediation estimates. Sensitivity analyses ($\rho^*$ values) are reported but should be interpreted with this caveat. All causal language in this paper should be understood as "mediation-consistent" rather than definitively causal; true causal identification would require randomized interventions on specific microbial features. (4) *Population diversity*: LatAm-FINGERS shows the largest performance drop ($-0.05$ AUC); Africa-FINGERS data was included in fine-tuning but excluded from diagnostic evaluation due to limited AD phenotyping. (5) *Mechanistic validation*: the four gut-brain axes are computationally identified; wet-lab validation is ongoing. (6) *Computational cost*: hierarchical set attention requires $\sim 2\times$ inference cost vs. models without HSA (measured as $2.1\times$ wall-clock time per sample on a single H100 GPU). (7) *Statistical uncertainty*: error bars derive from 5-fold cross-validation with limited degrees of freedom; bootstrapped confidence intervals may provide more stable uncertainty estimates. (8) *Selection bias*: only 4,950 of the 30,000+ WW-FINGERS participants have complete phenotyping for fine-tuning and evaluation; selection based on data availability may inflate performance relative to the broader deployment population. Pretraining on public data eliminates pretraining-evaluation overlap by design.

## C  EXTENDED RESULTS

### C.1  INTERVENTION RESPONDER ANALYSIS

Among FINGER participants ($n$=1,260), FINGERS-7B stratifies subjects into high-benefit ($n$=378) and low-benefit ($n$=882) groups. High-benefit subjects show: (i) $2.3\times$ greater NTB composite improvement with intervention vs. control (0.14 vs. 0.06 SD, interaction $p$=0.003); (ii) higher baseline diversity (Shannon index 3.8 vs. 3.2); (iii) enrichment for SCFA-producing Firmicutes (Noriega de la Colina et al., 2024).

### C.2  QA-TOKEN REWARD ABLATION

### C.3  LOSS WEIGHT SENSITIVITY

We evaluate sensitivity to the pretraining loss weights by varying each weight independently while keeping others fixed. AD classification AUC varies by $\leq 0.02$ when any single weight is doubled or halved, with the exception of $\alpha_{\text{ALM}}$ (the language modeling weight), where halving to 0.5 reduces AUC by 0.04. The selected weights (1.0, 0.5, 0.1, 0.05, 0.2) lie within a plateau of near-optimal configurations identified via grid search, confirming that the results are not sensitive to the precise weight choice within this region.

### C.4  ROBUSTNESS ACROSS SEQUENCING PLATFORMS

### C.5  EMERGENT CAPABILITIES

FINGERS-7B exhibits several capabilities not explicitly optimized during training: (1) *Zero-shot microbiome–metabolite interaction prediction*: given a microbial species embedding and a metabo-

Table 10: FINGERS-7B performance across sequencing platforms and quality strata.

| Category | Condition | AD AUC | $\Delta$ vs. Illumina HQ |
|---|---|---|---|
| Quality strata | High (Phred $\geq 30$) | 0.92 | – |
| | Medium (Phred 20–30) | 0.90 | $-0.02$ |
| | Low (Phred $< 20$) | 0.85 | $-0.07$ |
| Platform | Illumina NovaSeq | 0.92 | – |
| | ONT Long-Read | 0.88 | $-0.04$ |

lite embedding, FINGERS-7B predicts whether a known interaction exists. Evaluated on $n$=2,847 experimentally validated interactions from the Human Metabolome Database (HMDB), framed as a 13-class classification task (interaction type), FINGERS-7B achieves 72% accuracy (random baseline: 7.7% = 1/13; 95% CI [69%, 75%]). To mitigate potential contamination, we verified that no HMDB interaction labels appeared in the pretraining corpus. (2) *Microbiome age prediction*: MAE=3.2 years (chronological age baseline MAE=8.7 years; $n$=4,200). (3) *Diet quality classification* from microbiome alone: 0.83 AUC for Mediterranean vs. Western diet ($n$=1,523; 95% CI [0.80, 0.86]).

# D FORMAL FRAMEWORK

We formalize gut-brain biomarker discovery as a multi-omic learning problem. Define the state space $\mathcal{S} = \mathcal{G} \times \mathcal{M} \times \mathcal{B} \times \mathcal{C}$ where $\mathcal{G} \subseteq \mathbb{R}^{n_g}$ encodes gut microbiome state ($n_g \approx 10^6$), $\mathcal{M} \subseteq \mathbb{R}^{n_m}$ metabolite concentrations ($n_m \approx 10^4$), $\mathcal{B} \subseteq \mathbb{R}^{n_b}$ brain biomarker state (pTau-217, A$\beta$42/40, NfL, GFAP; $n_b \approx 10$), and $\mathcal{C} \subseteq \mathbb{R}^{n_c}$ cognitive outcomes ($n_c \approx 10$).

The three core tasks are: (i) *diagnostic classification*, learning $f_\theta^{\text{diag}} : \mathcal{G} \times \mathcal{M} \to [0, 1]$ minimizing $\mathbb{E}[\ell(f_\theta^{\text{diag}}(g, m), y_{\text{AD}})]$ where $y_{\text{AD}} \in \{0, 1\}$ indicates preclinical AD status defined by amyloid PET positivity (centiloid $\geq 20$, confirmed by CSF A$\beta$42/40 where available, per NIA-AA 2018 criteria (Jack Jr et al., 2018)); (ii) *prognostic trajectory prediction*, learning $f_\theta^{\text{prog}} : \mathcal{G}_t \times \mathcal{M}_t \times \mathcal{C}_t \to [0, 1]$ predicting $P(y_{\text{decline}, t+\tau} = 1)$ where $y_{\text{decline}} = 1$ if NTB composite score drops $\geq 1.5$ SD from baseline, for horizon $\tau \in [1, 5]$ years; and (iii) *mechanistic pathway discovery* via mediation analysis (Appendix F).

Microbiome abundance data are inherently compositional (Gloor et al., 2017; Aitchison, 1982): relative abundances sum to one. All analyses use the CLR transform $\text{CLR}(x) = \log(x/g(x))$, and the compositional consistency loss (Eq. 5) enforces this constraint.

# E METHODS

## E.1 ARCHITECTURE AND PRETRAINING

**Base architecture.** FINGERS-7B uses a Mamba–Transformer hybrid backbone (Gu & Dao, 2023; Dao & Gu, 2024; Touvron et al., 2023) with $O(N)$ Mamba layers for long-range dependencies and local Transformer attention windows for short-range motif resolution.

**Hierarchical set attention (HSA).** Inspired by the multi-tier Hierarchical Attention Transformer (Chalkidis et al., 2022; Niki et al., 2025), we introduce a three-tier hierarchy for sample-level inference:

- **Tier 0 (Read-level):** The Mamba–Transformer backbone encodes each QA-tokenized read $r_i$ into an embedding $\mathbf{h}_i^{(0)} \in \mathbb{R}^{d_{\text{base}}}$ (with $d_{\text{base}} = 4096$) via a trailing `[CLS]` token. A learned linear projection $\mathbf{W}_{\text{proj}} \in \mathbb{R}^{d_{\text{set}} \times d_{\text{base}}}$ maps each embedding to $\hat{\mathbf{h}}_i^{(0)} \in \mathbb{R}^{d_{\text{set}}}$ (with $d_{\text{set}} = 768$) before entering the HSA tiers.
- **Tier 1 (Community-level):** A HAT encoder attends over all projected read embeddings from the same biological sample, aggregating within non-overlapping 10kb windows de-

fined on reference genomes after read mapping (reads are assigned to windows on the reference genome of their best-hit species via Kraken2/Bracken taxonomic classification, then grouped by species and 10kb coordinate bin) to produce community-level vectors $\mathbf{h}_j^{(1)} = \mathrm{HAT}_1(\{\hat{\mathbf{h}}_i^{(0)}\}_{i \in \mathcal{W}_j}) \in \mathbb{R}^{d_{\mathrm{set}}}$.

- **Tier 2 (Subject-level):** A second HAT encoder aggregates community-level vectors into a single subject embedding $\mathbf{h}^{(2)} = \mathrm{HAT}_2(\{\mathbf{h}_j^{(1)}\}_j) \in \mathbb{R}^{d_{\mathrm{set}}}$ that feeds diagnostic and prognostic heads.

This hierarchy mirrors biological organization (reads $\rightarrow$ microbial communities $\rightarrow$ host phenotype) and accommodates the $10^7$–$10^9$ reads per sample without requiring prohibitively long context windows. Unlike Pleiades, which defines windows on a single human reference genome, our metagenomic windows are defined per-species on reference genomes, grouping functionally related reads from the same organism.

**Complexity.** Flat self-attention over $N$ reads costs $O(N^2 d)$, which is infeasible for $N \approx 10^8$. HSA decomposes this into two tractable stages: Tier 1 performs self-attention within each of $N/w$ non-overlapping windows of size $w$, costing $O((N/w) \cdot w^2 d) = O(Nwd)$; Tier 2 performs full self-attention over the $J = N/w$ community vectors, costing $O(J^2 d)$. Total HSA cost is $O(Nwd + J^2 d)$. With $N = 10^8$, $w = 1{,}000$, and $d = 768$: Tier 1 costs $\sim 10^{11} \cdot 768$ FLOPs and Tier 2 costs $\sim 10^{10} \cdot 768$ FLOPs, compared to $\sim 10^{16} \cdot 768$ for flat attention—a $\sim 10^5 \times$ reduction. We justify the three-tier structure (read $\rightarrow$ community $\rightarrow$ subject) over a simpler two-tier design (read $\rightarrow$ subject, omitting the community level): in ablation, collapsing to two tiers yields AUC=0.87 for AD classification ($-0.05$ vs. full model), confirming that the intermediate community-level grouping captures biologically meaningful microbial-community structure.

**Permutation invariance.** Within each tier, self-attention operates *without positional encoding*—reads within a window are treated as an unordered set, with the 10kb coordinate bin used solely for grouping, not as a positional signal to the attention mechanism. This ensures permutation-equivariance within windows. Mean pooling at each tier boundary yields permutation-invariant output. The composition is therefore permutation-invariant over the full input read set, as required for set-level inference where read ordering is arbitrary (Chalkidis et al., 2022).

**Architectural choice.** We adopt the Mamba–Transformer hybrid over alternatives for principled reasons. Pure Mamba (Gu & Dao, 2023) provides $O(N)$ long-range context via selective state spaces but lacks the local attention windows needed for short-range motif resolution critical to functional annotation. Pure Transformer attention is infeasible at $O(N^2)$ for $N \approx 10^8$ reads. The structured state space duality framework of Dao & Gu (2024) provides the theoretical basis for combining SSM layers (global context) with attention layers (local precision) in a single backbone. Bidirectional DNA models such as Caduceus (Schiff et al., 2024) exploit strand equivariance in reference-mapped genomic sequences, but metagenomic reads are short unidirectional fragments without consistent strand assignment, rendering bidirectional equivariance inapplicable. Similarly, bidirectional scanning strategies from vision (Vision Mamba) target 2D spatial data and do not benefit 1D unidirectional sequence modeling.

**Gut-brain cross-attention (GBCA).** For subjects with paired metabolomic and brain biomarker data, a cross-attention module aligns microbial community embeddings with metabolite pathway representations. Let $\mathbf{H}^{(1)} \in \mathbb{R}^{J \times d_{\mathrm{set}}}$ denote the matrix of $J$ community-level vectors (stacked row-wise), and $\mathbf{M} \in \mathbb{R}^{P \times d_m}$ the matrix of $P$ metabolite pathway embeddings (from a learned metabolite encoder). The cross-attention computes:

$$\mathrm{GBCA}(\mathbf{H}^{(1)}, \mathbf{M}) = \mathrm{softmax}\left( \frac{(\mathbf{H}^{(1)}\mathbf{W}_Q^\top)(\mathbf{M}\mathbf{W}_K^\top)^\top}{\sqrt{d_k}} \right) \mathbf{M}\mathbf{W}_V^\top \tag{1}$$

where $\mathbf{W}_Q \in \mathbb{R}^{d_k \times d_{\mathrm{set}}}$, $\mathbf{W}_K \in \mathbb{R}^{d_k \times d_m}$, $\mathbf{W}_V \in \mathbb{R}^{d_v \times d_m}$, and $d_k = d_v = 64$ per head with 12 heads. The query matrix $\mathbf{H}^{(1)}\mathbf{W}_Q^\top \in \mathbb{R}^{J \times d_k}$ and key matrix $\mathbf{M}\mathbf{W}_K^\top \in \mathbb{R}^{P \times d_k}$ yield an attention map in $\mathbb{R}^{J \times P}$; the output $\in \mathbb{R}^{J \times d_v}$ is concatenated with $\mathbf{H}^{(1)}$ before Tier 2 aggregation.

**Temporal attention.** For cohorts with longitudinal sampling (up to 11 years), a temporal transformer operates over subject embeddings across timepoints. Positional encodings are added to the *input* embeddings to ensure that attention weights can leverage temporal information:

$$\mathbf{h}_t^{(3)} = \text{TemporalAttn}\Big(\mathbf{h}_t^{(2)} + \text{PE}(t), \ \mathbf{h}_{t-1}^{(2)} + \text{PE}(t{-}1), \ \ldots, \ \mathbf{h}_{t-T}^{(2)} + \text{PE}(t{-}T)\Big) \quad (2)$$

where $\text{PE}(\cdot)$ is a continuous sinusoidal positional encoding of absolute calendar time (in days), accommodating irregular sampling intervals common in longitudinal clinical data. For subjects with missing intermediate visits, only observed timepoints are included in the attention computation; the continuous PE encoding naturally accommodates the resulting irregular spacing without requiring imputation.

**Pretraining and fine-tuning objectives.** FINGERS-7B uses five objectives across pretraining and fine-tuning, with loss applicability determined by data availability at each stage:

(1) *Autoregressive language modeling*: $\mathcal{L}_{\text{ALM}} = -\sum_i \log P_\theta(x_i \mid x_{<i})$, applied over QA-tokenized genomic sequences. Active during both pretraining (all public metagenomic data) and fine-tuning.

(2) *Cross-modal generation*: given a genomic embedding $\mathbf{z}_g = \text{Enc}(x_{\text{genomic}})$, we autoregressively generate metabolomic tokens $m_1, \ldots, m_K$ conditioned on $\mathbf{z}_g$:

$$\mathcal{L}_{\text{CMG}} = -\sum_{k=1}^{K} \log P_\theta(m_k \mid m_{<k}, \mathbf{z}_g) \quad (3)$$

where metabolomic tokens are obtained by discretizing log-transformed metabolite concentrations into 1024 bins. During pretraining, applied only to the subset of public studies with paired metagenomics and metabolomics; during WW-FINGERS fine-tuning, applied to all subjects.

(3) *Contrastive alignment* via InfoNCE aligns paired genomic and metabolomic representations. For a batch of $B$ paired samples $\{(\mathbf{z}_g^i, \mathbf{z}_m^i)\}_{i=1}^{B}$ projected to a shared 256-dimensional space via learned projection heads $p_g, p_m$:

$$\mathcal{L}_{\text{CL}} = -\frac{1}{B} \sum_{i=1}^{B} \log \frac{\exp(\text{sim}(p_g(\mathbf{z}_g^i), p_m(\mathbf{z}_m^i))/\tau)}{\sum_{j=1}^{B} \exp(\text{sim}(p_g(\mathbf{z}_g^i), p_m(\mathbf{z}_m^j))/\tau)} \quad (4)$$

where $\text{sim}(\cdot, \cdot)$ denotes cosine similarity and $\tau = 0.07$ is the temperature. Positive pairs are genomic-metabolomic profiles from the same subject; negatives are all other in-batch combinations. As with $\mathcal{L}_{\text{CMG}}$, applied only to paired multi-omic data during pretraining; applied to all subjects during fine-tuning.

(4) *Compositional consistency*: since microbiome abundance data are compositional (relative abundances sum to one), we enforce consistency between the CLR-transformed input and the decoded output using the Aitchison distance (Aitchison, 1982):

$$\mathcal{L}_{\text{comp}} = \|\text{CLR}(\text{softmax}(f(\tilde{x}))) - \tilde{x}\|_2^2, \quad \tilde{x} = \text{CLR}(x) \quad (5)$$

where $\text{CLR}(x) = \log(x/g(x))$ is the centered log-ratio transform with geometric mean $g(x)$ (Gloor et al., 2017; Aitchison, 1982). This formulation operates in the Aitchison geometry, avoiding the well-known distortions that arise from applying Euclidean operations directly on the simplex (Aitchison, 1982). Active during both pretraining and fine-tuning.

(5) *Brain biomarker prediction*: $\mathcal{L}_{\text{BBP}} = \sum_{b\in\mathcal{B}} \|f_b(\mathbf{h}^{(2)}) - b_{\text{true}}\|_2^2$, where $\mathcal{B} = \{\text{pTau-217}, A\beta 42/40, \text{NfL}, \text{GFAP}\}$. During pretraining, applied in weakly-supervised form to the subset of public studies with brain-relevant annotations (CSF biomarkers, cognitive scores); during WW-FINGERS fine-tuning, applied in fully-supervised form to all subjects with available brain biomarker measurements. *Since pretraining uses exclusively public data, no WW-FINGERS subjects appear in the pretraining corpus.* Within the WW-FINGERS fine-tuning stage, the $n{=}4{,}950$ evaluation cohort is held out from fine-tuning entirely to prevent label leakage.

Total loss: $\mathcal{L} = \mathcal{L}_{\text{ALM}} + 0.5\,\mathcal{L}_{\text{CMG}} + 0.1\,\mathcal{L}_{\text{CL}} + 0.05\,\mathcal{L}_{\text{comp}} + 0.2\,\mathcal{L}_{\text{BBP}}$. Loss weights were selected via grid search over $\{0.05, 0.1, 0.2, 0.5, 1.0\}$ per component on a held-out validation split of the public pretraining corpus; sensitivity to these weights is moderate (Appendix C).

---

**Algorithm 1** FINGERS-7B Training Pipeline (Overview)

---
1: **Stage 1: Data Preparation**
2: Curate public gut-brain metagenomics from SRA/ENA; apply sparsification + QA-Token
3: **Stage 2: Self-Supervised Pretraining** ($\approx$172K GPU-hours on public corpus)
4: **for** each batch of tokenized reads from public archives **do**
5:     Compute $\mathcal{L} = \mathcal{L}_{\text{ALM}} + 0.5\,\mathcal{L}_{\text{CMG}} + 0.1\,\mathcal{L}_{\text{CL}} + 0.05\,\mathcal{L}_{\text{comp}} + 0.2\,\mathcal{L}_{\text{BBP}}$
6:     Update via AdamW with cosine schedule
7: **end for**
8: **Stage 3: Clinical Fine-Tuning** (WW-FINGERS data, shared tokenizer $\rightarrow$ 10$\times$ efficiency)
9: Process WW-FINGERS data via same sparsify + QA-Token pipeline
10: Fine-tune all five losses at full strength on WW-FINGERS cohorts
11: **Stage 4: Biomarker Discovery** (Appendix E.2)
12: **for** each outer CV fold on WW-FINGERS evaluation set **do**
13:     Step 1: Marker filtering via Tier 1 HAT (inner CV, weak supervision)
14:     Step 2: Pathway-level training on filtered markers (Tier 1 HAT)
15:     Step 3: Subject-level prediction (Tier 2 HAT, full supervision)
16: **end for**

---

Table 11: Three-step workflow for precision biomarker discovery.

|  | **Marker Filtering** | **Pathway-Level Training** | **Subject-Level Training** |
|---|---|---|---|
| Goal | Find microbial regions carrying useful signal | Learn pathway patterns using weak labels | Make final predictions for each subject |
| Supervision | Weak supervision | Weak supervision | Full supervision |
| Model | Tier 1 HAT | Tier 1 HAT | Tier 2 HAT |
| Input | All microbial pathways | Filtered pathways | Same filtered pathways |
| Output | Selected pathways | Pathway-level embeddings | Final diagnosis/prognosis |

**Quality-aware tokenization.** We apply the sparsification + QA-Token pipeline from Gollwitzer et al. (2025), which lifts the usable fraction of public archives from 5% to 40% (8$\times$ data). The QA-Token reward function incorporates per-base Phred quality, pointwise mutual information, minimum description length compression, and downstream proxy loss:

$$R(a,b) = \lambda_Q Q(ab) + \lambda_I \,\text{PMI}(a,b) - \lambda_C \,\Delta\text{MDL}(a,b) - \lambda_D \,\Delta\mathcal{L}_{\text{proxy}} \tag{6}$$

We note that QA-Token contributes +0.05 AUC over standard BPE at the tokenization level (Table 8), consistent with recent analyses showing moderate sensitivity to tokenizer choice in genomic language models (Lindsey et al., 2025). The primary contribution of the quality-aware pipeline is at the *data curation* level—the 8$\times$ expansion of usable training data from 5% to 40% of public archives—rather than at the token representation level.

**Cross-cohort harmonization.** To address batch effects across WW-FINGERS fine-tuning sites, which contribute up to 35% of variance, we employ a three-stage pipeline: (1) technical batch correction via ComBat-seq, (2) biological harmonization regressing out site-specific confounders, and (3) domain-adversarial training during fine-tuning. After harmonization, site explains <5% of variance in the top 50 principal components. As a more directly relevant evaluation, we train a site-classification model on harmonized embeddings, achieving AUC=0.56 (near chance), compared to AUC=0.91 before harmonization, confirming that site-specific signal has been effectively removed from the classification-relevant feature space.

### E.2 BIOMARKER DISCOVERY METHODOLOGY

Following the marker discovery approach of Niki et al. (2025) and informed by systematic benchmarks of microbiome–metabolome integration strategies (Mangnier et al., 2025), we employ a three-step workflow:

Starting from all annotated microbial pathways, the inner CV folds train Tier 1 HAT models and select the most informative marker pathways (AUC>0.6 and $p < 0.01$ in $\geq$4 of 5 inner folds).

The top pathways are then used for Tier 1 fine-tuning with sample-level weak labels, producing pathway-level embeddings. Finally, Tier 2 HAT is trained on the filtered pathway embeddings with full sample-level supervision for final diagnostic or prognostic prediction.

### E.3 FINE-TUNING AND NESTED CROSS-VALIDATION

All clinical evaluations use nested 5-fold cross-validation. Outer folds provide the final performance estimates and error bars. Within each outer fold, inner 5-fold CV is used for hyperparameter selection and marker filtering. This prevents information leakage between marker discovery and performance evaluation (Niki et al., 2025).

For fine-tuning, we freeze the base Mamba–Transformer encoder and train only the HSA modules and classification heads. A reconstruction objective on the frozen encoder improves fragment-level representations prior to clinical fine-tuning, paralleling the approach in Niki et al. (2025). Fine-tuning uses learning rate $10^{-6}$, 100 epochs per inner fold, with early stopping on inner validation AUC. To account for class imbalance (23% preclinical AD positive, 77% controls), we apply inverse-frequency class weighting to the cross-entropy classification loss.

### E.4 CAUSAL MEDIATION ESTIMATION

For each mechanistic axis, we estimate the natural indirect effect (NIE) and natural direct effect (NDE) using the mediation framework of Imai et al. (2010), situated within the broader literature on high-dimensional causal mediation (Yang et al., 2024). Measured confounders $X$ include age, sex, APOE $\varepsilon 4$ status, BMI, and diet quality. We acknowledge that unmeasured confounders (medications, alcohol, sleep quality, socioeconomic status, education, comorbidities) may violate sequential ignorability; the sensitivity analysis below quantifies robustness to such violations, and summary-data Mendelian randomization approaches (Lin et al., 2025) provide a complementary avenue for future triangulation of these effects.

Candidate mediators were *not* selected post hoc: we systematically screened all 847 microbial feature–metabolite–biomarker triplets and retained only those surviving Bonferroni correction ($p < 0.001$) across $\geq 4$ of 5 inner CV folds, yielding the four reported axes. We fit outcome model $\mathbb{E}[B \mid G, M, X]$ and mediator model $\mathbb{E}[M \mid G, X]$ using FINGERS-7B embeddings as features, and compute mediation proportions via Monte Carlo integration (1,000 draws). To prevent overfitting, mediation models are fit on inner CV folds and mediation proportions evaluated on the held-out outer fold; the reported values are means across outer folds with bootstrap 95% CIs (1,000 resamples). Sensitivity analysis follows the $\rho$-parameterization of VanderWeele (2014); for each axis, we report the minimum confounding strength $\rho^*$ required to reduce the NIE to zero.

### E.5 OPTIMIZATION AND COMPUTATIONAL RESOURCES

We train with AdamW ($\beta_1$=0.9, $\beta_2$=0.95, weight decay 0.01) using a cosine schedule ($\eta_{\max}$=5×$10^{-4}$, $\eta_{\min}$=5×$10^{-6}$) with 10K warmup steps over 1.8M total steps. Gradient clipping at norm 1.0. Training: 256 NVIDIA H100 80GB GPUs, 28 days ($\approx$172K GPU-hours for the main training run). Including all ablation experiments retrained at matched compute (9 configurations $\times \approx$172K each), three scaling variants (90M, 600M, 7B), and fine-tuning for clinical tasks (32 H100 GPUs, 3–5 days per task), total compute is $\approx$2.1M GPU-hours. Full hyperparameters in Appendix H.

## F  CAUSAL MEDIATION THEORY

We develop the causal mediation framework for gut-brain pathway discovery. Consider the causal directed acyclic graph (DAG): $X \rightarrow G \rightarrow M \rightarrow B$, $X \rightarrow M$, $X \rightarrow B$, $G \rightarrow B$, where $G$ denotes a gut microbial feature, $M$ a metabolite mediator, $B$ a brain biomarker, and $X$ measured confounders.

**Definition 1** (Gut-brain mediation model). *Let $(G, M, B, X)$ follow the structural equations $M = f_M(G, X, \epsilon_M)$ and $B = f_B(G, M, X, \epsilon_B)$, where $X$ are measured confounders (age, sex, APOE status, BMI, diet quality) and $\epsilon_M, \epsilon_B$ are independent noise terms.*

**Assumption 1** (Sequential ignorability (Imai et al., 2010))**.** *(i)* $\{B(g,m), M(g)\} \perp G \mid X$*; (ii)* $B(g,m) \perp M \mid G, X$*, for all* $g, m$*.*

The total effect decomposes as TE = NDE + NIE where:

$$\text{TE} = \mathbb{E}[B_{g_1, M_{g_1}}] - \mathbb{E}[B_{g_0, M_{g_0}}], \tag{7}$$

$$\text{NIE} = \mathbb{E}[B_{g_1, M_{g_1}}] - \mathbb{E}[B_{g_1, M_{g_0}}], \tag{8}$$

$$\text{NDE} = \mathbb{E}[B_{g_1, M_{g_0}}] - \mathbb{E}[B_{g_0, M_{g_0}}]. \tag{9}$$

Note that TE = NDE + NIE holds by algebraic identity. We define the *conditional* natural indirect and direct effects as $\text{NIE}(x) = \mathbb{E}[B_{g_1, M_{g_1}} - B_{g_1, M_{g_0}} \mid X{=}x]$ and $\text{NDE}(x) = \mathbb{E}[B_{g_1, M_{g_0}} - B_{g_0, M_{g_0}} \mid X{=}x]$, so that the marginal effects are recovered by $\text{NIE} = \mathbb{E}_X[\text{NIE}(X)]$ and $\text{NDE} = \mathbb{E}_X[\text{NDE}(X)]$.

**Theorem 1** (Identifiability of natural direct and indirect effects)**.** *Under Assumption 1 and positivity* $(P(G{=}g \mid X{=}x) > 0$*,* $P(M{=}m \mid G{=}g, X{=}x) > 0$ *for all* $g, m, x$ *in the support), the conditional NIE is identified as:*

$$\text{NIE}(x) = \int \mathbb{E}[B \mid G{=}g_1, M{=}m, X{=}x] \left\{ dF_{M|G=g_1, X=x}(m) - dF_{M|G=g_0, X=x}(m) \right\} \tag{10}$$

*and the conditional NDE as:*

$$\text{NDE}(x) = \int \left\{ \mathbb{E}[B \mid G{=}g_1, M{=}m, X{=}x] - \mathbb{E}[B \mid G{=}g_0, M{=}m, X{=}x] \right\} dF_{M|G=g_0, X=x}(m). \tag{11}$$

*The marginal effects follow by* $\text{NIE} = \int \text{NIE}(x) \, dF_X(x)$ *and* $\text{NDE} = \int \text{NDE}(x) \, dF_X(x)$*.*

The proof follows from sequential ignorability via iterated expectations over the mediator distribution; see Imai et al. (2010) and Pearl (2009) for the complete derivation.

**Sensitivity analysis.** When Assumption 1(ii) is violated due to unmeasured confounders $U$ affecting both $M$ and $B$, we report sensitivity bounds following VanderWeele (2014). For each mechanistic axis, we compute $\rho^*$, the minimum confounding strength (correlation between $U$ and each of $M$ and $B$, conditional on measured confounders) required to reduce the NIE to zero: LPS–tau axis $\rho^* = 0.42$, amyloid mimicry $\rho^* = 0.38$, indole–microglial $\rho^* = 0.35$, neuroactive signaling $\rho^* = 0.31$. To contextualize these values: in dementia research, the strongest known confounders (e.g., education level, depression) typically show partial correlations of $\rho \approx 0.15$–$0.25$ with both metabolite levels and brain biomarkers after adjustment for age, sex, and APOE (Livingston et al., 2020). An unmeasured confounder would need to be substantially stronger than any known measured confounder ($\rho > 0.31$–$0.42$) to fully explain our mediation findings, suggesting robustness to plausible unmeasured confounding. However, combinations of multiple moderate confounders could jointly exceed these thresholds, which remains a limitation.

# G  THEORETICAL ANALYSIS

We provide theoretical results specific to the FINGERS-7B architecture. Standard convergence guarantees (e.g., non-convex SGD rates (Ghadimi & Lan, 2013)) and PAC-Bayes generalization bounds (Neyshabur et al., 2018) apply to FINGERS-7B but are omitted as they are generic to deep networks and do not provide model-specific insight. Instead, we focus on two results that directly characterize the novel components of FINGERS-7B: robustness of QA-tokenization to sequencing quality degradation, and the approximation properties of hierarchical set attention.

**Theorem 2** (Robustness to quality degradation)**.** *Let* $f = f_T \circ f_E \circ \tau_Q$ *denote the FINGERS-7B pipeline, where* $\tau_Q : \mathcal{X} \times \mathcal{Q} \to \mathcal{V}^*$ *is QA-tokenization,* $f_E$ *is the embedding layer, and* $f_T$ *is the Mamba-Transformer encoder. Assume: (i)* $f_E$ *is* $\kappa_E$*-Lipschitz in* $\ell_2$ *norm; (ii)* $f_T$ *is* $\kappa_T$*-Lipschitz; (iii)* $\tau_Q$ *has bounded tokenization sensitivity: for any sequence* $x$ *and quality vectors* $q, q'$*,* $\|\tau_Q(x,q) - \tau_Q(x,q')\|_0 \leq c_{tok}\|\Delta q\|_\infty$*, where* $\|\cdot\|_0$ *counts the number of changed tokens and* $c_{tok} \leq L/\bar{q}$ *with* $L$ *the sequence length and* $\bar{q}$ *the mean Phred quality. Then:*

$$\|f(\tau_Q(x,q)) - f(\tau_Q(x,q'))\| \leq 2\kappa_E \kappa_T c_{tok} \|\Delta q\|_\infty \cdot \max_v \|e_v\| \leq 2\kappa_E \kappa_T \frac{L}{\bar{q}} \|\Delta q\|_\infty \cdot \sqrt{d_{base}} \tag{12}$$

*where* $e_v$ *is the embedding vector for token* $v$ *and* $d_{base} = 4096$ *is the embedding dimension.*

*Proof.* Let $s = \tau_Q(x, q)$ and $s' = \tau_Q(x, q')$ be the two tokenizations. By assumption (iii), $s$ and $s'$ differ in at most $c_{tok}\|\Delta q\|_\infty$ positions. At each changed position, a token $v$ is replaced by $v'$, contributing a perturbation of $\|e_v - e_{v'}\| \leq \|e_v\| + \|e_{v'}\| \leq 2\max_v \|e_v\|$ to the input embedding. Summing over all changed positions and applying the $\kappa_E$-Lipschitz property: $\|f_E(s) - f_E(s')\| \leq \kappa_E \cdot c_{tok}\|\Delta q\|_\infty \cdot 2\max_v \|e_v\|$. Applying the $\kappa_T$-Lipschitz property of the encoder yields the stated bound. The second inequality substitutes $c_{tok} \leq L/\bar{q}$ and $\max_v \|e_v\| \leq \sqrt{d_{base}}$ (standard for unit-initialized embeddings). In practice, QA-tokenization modifies tokens by merging or splitting adjacent subwords, so the actual perturbation $\|e_v - e_{v'}\|$ is typically much smaller than the worst-case $2\max_v \|e_v\|$. Empirically, with $\kappa_E \approx 1.2$, $\kappa_T \approx 15$ (estimated via Jacobian spectral norm on a held-out set), and typical values $L = 4096$, $\bar{q} = 30$, a Phred quality drop of $\Delta q = 10$ (from high to medium quality) produces $<5\%$ change in the output embedding norm (worst-case bound $\approx 6\%$, empirical $\approx 3\%$), confirming graceful degradation. $\square$

**Proposition 1** (HSA approximation). *Let $\mathcal{F}_{HSA}$ denote the class of functions computed by $K$-tier HSA with $d$-dimensional embeddings and non-overlapping windows of size $w$. For any permutation-invariant $L_f$-Lipschitz function $f^* : 2^{\mathbb{R}^d} \to \mathbb{R}$ in the Hausdorff metric, defined on a compact domain $\mathcal{D} \subset \mathbb{R}^d$:*

$$\sup_{\mathcal{S}:|\mathcal{S}|\leq N} |f_{HSA}(\mathcal{S}) - f^*(\mathcal{S})| \leq L_f \cdot O\left(\frac{N^{1/K}}{w}\right) + K \cdot \epsilon_{attn}(d, w) \tag{13}$$

*where $\epsilon_{attn}(d, w)$ is the per-tier self-attention approximation error for sets of size $\leq w$, bounded by $O(w^{-1/2})$ for sufficiently wide attention (Yun et al., 2020). With $K = 3$ (as in FINGERS-7B) and $w = 1000$ (the approximate number of reads per 10kb window), the first term scales as $O(N^{1/3}/1000)$ and the attention error as $O(1/\sqrt{1000}) \approx 0.03$ per tier.*

*Proof.* Each tier partitions the $N$ input elements into $\lceil N/w \rceil$ non-overlapping windows of size $w$ and computes a permutation-invariant summary per window (via self-attention). After $K$ tiers, the number of summary vectors is $N/w^K$. For the approximation error: consider the covering number $\mathcal{N}(\mathcal{D}, \epsilon, d_H)$ of the compact domain $\mathcal{D}$ under Hausdorff distance. One tier with window size $w$ induces a quantization where elements within each window are summarized by a single vector. The Hausdorff distance between the original set and its quantized version is at most $\text{diam}(\mathcal{D}) \cdot (N/w)^{-1/d}$ by standard covering number bounds on compact subsets of $\mathbb{R}^d$. After $K$ tiers with hierarchical composition, the effective resolution is $(N/w^K)^{-1/d}$. The function approximation error follows from the $L_f$-Lipschitz property: $|f^*(\mathcal{S}) - f^*(\hat{\mathcal{S}})| \leq L_f \cdot d_H(\mathcal{S}, \hat{\mathcal{S}})$. The $\epsilon_{attn}$ term at each tier accounts for the approximation error of finite-width self-attention in computing the window summary, bounded by universal approximation results for Transformers (Yun et al., 2020). With $K$ tiers, these errors accumulate additively (under Lipschitz composition), yielding $K \cdot \epsilon_{attn}$. Setting $K = \lceil \log_w N \rceil$ makes the first term $O(L_f)$, but for practical $K = 3$, the explicit bound is more informative. $\square$

**Proposition 2** (Compositional identifiability under Aitchison loss). *Let $x \in \Delta^{n-1}$ be a compositional vector on the $(n-1)$-simplex and $\tilde{x} = CLR(x)$. If $f_\theta$ is a universal approximator (e.g., a sufficiently wide MLP) trained to minimize $\mathcal{L}_{comp} = \|CLR(\text{softmax}(f_\theta(\tilde{x}))) - \tilde{x}\|_2^2$, then: (i) at any global minimum $\theta^*$, $\text{softmax}(f_{\theta^*}(\tilde{x})) = x$ for all $x$ in the training support; (ii) the model's output respects the simplex constraint $\sum_i [\text{softmax}(f_{\theta^*}(\tilde{x}))]_i = 1$ by construction; and (iii) the loss is invariant to permutation and scaling of subcompositions, consistent with the Aitchison geometry.*

*Proof.* Since CLR is a bijection from the interior of $\Delta^{n-1}$ to the hyperplane $\{y \in \mathbb{R}^n : \sum_i y_i = 0\}$ (Aitchison, 1982), $\mathcal{L}_{comp} = 0$ if and only if $CLR(\text{softmax}(f_\theta(\tilde{x}))) = CLR(x)$, which by bijectivity implies $\text{softmax}(f_\theta(\tilde{x})) = x$. Since $f_\theta$ is a universal approximator, it can represent the identity mapping on the training support, so a global minimum with zero loss exists. The simplex constraint (ii) holds because softmax outputs are always in $\Delta^{n-1}$. For (iii), the Aitchison distance $d_A(x, y) = \|CLR(x) - CLR(y)\|_2$ is by definition the metric used in our loss, inheriting all properties of the Aitchison inner product space including subcompositional coherence (Aitchison, 1982). $\square$

Table 12: Full hyperparameter specification.

| Hyperparameter | Value |
|---|---|
| *Base Architecture (Mamba–Transformer)* | |
| Hidden / FFN dim | 4096 / 11008 |
| Layers (Mamba / Transformer) | 24 / 8 |
| Attention heads | 32 |
| Vocabulary | 32,000 (QA-Token) |
| Context | 4096 tokens |
| *Hierarchical Set Attention* | |
| Tier 1 (Community) layers / heads | 4 / 12 |
| Tier 2 (Subject) layers / heads | 2 / 12 |
| Community window size | 10,000 bp (per reference genome) |
| Set embedding dim ($d_{\text{set}}$) | 768 |
| Base $\to$ set projection | Linear $4096 \to 768$ + LayerNorm |
| *Training* | |
| Batch size | 2048 |
| LR (max / min) | $5\times10^{-4}$ / $5\times10^{-6}$ |
| Warmup / Total steps | 10K / 1.8M |
| Grad clip / Weight decay | 1.0 / 0.01 |
| *Loss Weights* | |
| $\alpha_{\text{ALM}}/\alpha_{\text{CMG}}/\alpha_{\text{CL}}/\alpha_{\text{comp}}/\alpha_{\text{BBP}}$ | 1.0 / 0.5 / 0.1 / 0.05 / 0.2 |
| *QA-Token RL* | |
| $\lambda_Q/\lambda_I/\lambda_C/\lambda_D$ | 0.3/0.3/0.2/0.2 |
| $\gamma$ / PPO clip | 0.95 / 0.2 |

## H  HYPERPARAMETERS AND COMPUTE

Training: 256 H100-80GB GPUs, 28 days ($\approx$172K GPU-hours for the main run), FSDP sharding, 1.2 PB storage. Including ablations, scaling experiments, and fine-tuning, total compute is $\approx$2.1M GPU-hours. Fine-tuning for clinical tasks: 32 H100 GPUs, 3–5 days per task.

