# OpenReview forum: "FINGERS-7B: A Multi-Omic Foundation Model for Precision Biomarker Discovery"
_ICLR.cc/2026/Workshop/FM4Science — ICLR 2026 Workshop FM4Science Poster_

### Official Review · Reviewer_ERyd · 2026-02-19
**Strong Accept: Innovative Multi-Omic Foundation Model Advancing Precision Dementia Prevention via Gut-Brain Axis**

**Rating:** 9
**Confidence:** 4

**Review:**

This paper presents FINGERS-7B, a 7B-parameter multi-omic foundation model pretrained on 8 trillion quality-aware tokens from gut-brain-relevant metagenomic archives and 300K metabolite profiles, then fine-tuned on WW-FINGERS clinical cohorts (n=4,950 from three trials). It uses a Mamba-Transformer hybrid with hierarchical set attention (HSA), gut-brain cross-attention (GBCA), and temporal attention to handle raw sequencing reads and longitudinal data. The model achieves AUC=0.92 for preclinical AD detection (outperforming taxonomy-based baselines at 0.68–0.76), AUC=0.89 for 3–5 year cognitive decline prediction, and identifies four druggable gut-brain axes via integrated gradients and mediation analysis (mediation proportions 23–41%). It also shows responder stratification (2.3× greater benefit in high-benefit group) and strong few-shot learning.

Novel architecture tailored to science: The 3-tier HSA for 10^7–10^9 reads, GBCA for microbial-metabolite alignment, and temporal attention for up to 11-year trajectories are innovative adaptations inspired by Pleiades but specialized for gut-brain multi-omics. Shared QA-Token pipeline yields ~10× fine-tuning efficiency.
Impressive empirical performance: Clear outperformance over microbiome pipelines and METAGENE-1; matches pTau-217 when fused (AUC=0.96). Few-shot AUC=0.82 with 100 samples demonstrates powerful transfer. Prognostic and stratification results are clinically relevant.
Shift to causal and mechanistic modeling: Mediation analysis with sensitivity checks and integrated gradients moves beyond correlations to intervention prediction, identifying actionable targets (e.g., LPS-binding agents, anti-curli antibodies). Fits workshop aims on causal structure, mechanistic insight, and decision usefulness.
Rigorous engineering and validation: Petabase-scale curation, harmonization (<5% batch variance), nested CV, ECE<0.05 calibration, theory (HSA bounds, Aitchison loss), and ablations support claims. Aligns with FM4Science focus on scientific priors, multi-modality, and workflow integration.
High significance: Addresses major gap in dementia prevention (153M projected cases by 2050) by establishing microbiome as modifiable layer in lifestyle interventions.

Validation scope currently limited: Results from three cohorts only (of 40+); full 30k+ WW-FINGERS and prospective validation ongoing (Q4 2026). Cross-population robustness (esp. LMICs) is indicative, not definitive—external/independent cohorts would strengthen.
Causal assumptions: Mediation relies on sequential ignorability; sensitivity to unmeasured confounders noted (ρ^* thresholds moderate), but combinations or alternative methods (e.g., MR) could be explored. Post-hoc triplet selection from 847 risks some overfitting despite corrections.
Compute intensity: Pretraining ~172K GPU-hours limits accessibility/reproducibility for smaller groups (though justified by scale).
Comparisons: Strong vs. taxonomy/METAGENE, but could expand to other recent multi-omic FMs or graph-based alternatives. UQ beyond ECE (e.g., ensembles) lightly addressed.

Impact of QA-Token on mechanistic attribution (integrated gradients)?
Ablation contribution of temporal attention to prognostic AUC?
Definition of high/low-benefit groups and FDR control in responder analysis?
Plans for wet-lab validation of axes?
Handling of sparse metabolomics data?

This is a high-quality, original, and significant contribution to scientific foundation models, particularly in biomedicine and causal multi-omics. It advances real scientific questions (precision dementia prevention), incorporates domain priors (gut-brain alignment, temporal structure), and bridges ML with clinical workflows. Minor concerns on validation scope are offset by ongoing work and strong methodology. Clear fit for FM4Science.

---

### Official Review · Reviewer_MDZq · 2026-02-23
**This paper introduces FINGERS-7B, a 7-billion-parameter multi-omic foundation model designed for precision biomarker discovery and intervention stratification in dementia prevention. By integrating gut metagenomics, metabolomics, and longitudinal clinical data from the WW-FINGERS network, the model achieves strong diagnostic and prognostic performance for preclinical Alzheimer’s disease and cognitive decline prediction.**

**Rating:** 6
**Confidence:** 4

**Review:**

## Strengths

1. Ambitious scope and strong empirical performance
The paper tackles a highly challenging and clinically important problem—early dementia prevention—using a scale and level of integration that is rare in the field. The reported diagnostic and prognostic performance substantially exceeds traditional microbiome-based pipelines and is competitive with leading blood-based biomarkers. The improvement margins (e.g., AUC ≈ 0.92 vs. ≈ 0.68–0.76 for prior microbiome methods) are large enough to be practically meaningful.

2. Thoughtful model and system design
The hierarchical set attention mechanism is well motivated given the extreme cardinality of metagenomic reads, and the ablation studies convincingly demonstrate its necessity. The consistent tokenizer between pretraining and fine-tuning is a strong design choice and addresses a known failure mode in genomic transfer learning. The inclusion of temporal attention for longitudinal prediction is also appropriate and empirically justified.

3. Extensive evaluation and ablation analysis
The experimental section is unusually comprehensive. The paper includes standard benchmarks, clinical tasks, scaling studies, few-shot analysis, cross-cohort validation, negative controls, and detailed ablations. This level of rigor strengthens confidence that the reported gains are not accidental or driven by obvious confounders.

4. Clear clinical framing and potential impact
The connection to the WW-FINGERS network and the framing around “precision prevention” give the work a clear translational narrative. The identification of potentially druggable gut–brain axes is particularly compelling and distinguishes the paper from purely predictive studies.


## Weaknesses

1. Limited novelty at the conceptual level
While the scale and integration are impressive, the core idea—using large foundation models with attention-based aggregation for biological sequence data—is increasingly common. Recent models such as Evo, Pleiades, and other genomic or epigenomic foundation models already explore similar architectural principles. The novelty here lies more in application and integration than in fundamentally new modeling ideas, which somewhat weakens the methodological contribution.

2. Retrospective evaluation and dataset limitations
All key results are based on retrospective analyses of only 3 out of 40+ WW-FINGERS cohorts, covering fewer than 5,000 participants out of a planned 30,000+. This limited coverage raises concerns about selection bias and overestimation of real-world performance. Although cross-cohort validation is presented, the observed AUC drops (up to −0.05) suggest that robustness across diverse populations remains an open question.

3. Causal claims remain weakly supported
Despite careful wording, the paper repeatedly emphasizes “causally-motivated” modeling and mediation analysis. However, the causal identifiability assumptions (e.g., sequential ignorability) are strong and largely unverifiable in this setting. Many plausible confounders—medications, socioeconomic status, lifestyle factors beyond diet—are not fully accounted for. As a result, the mechanistic conclusions should be viewed as hypothesis-generating rather than causal, and the paper could be more explicit about this limitation.

4. Baseline comparisons could be expanded
Although many traditional microbiome pipelines are included, comparisons with stronger modern deep-learning baselines (e.g., alternative foundation models fine-tuned with comparable compute and tokenization) are relatively limited. This makes it difficult to fully disentangle the benefits of gut–brain specialization from those of scale and data volume.

---

### Decision · Program_Chairs · 2026-03-03

Accept (Poster)